# When majority rules, minority loses: bias amplification of gradient descent

## Abstract

Despite growing empirical evidence of bias amplification in machine learning, its theoretical foundations remain poorly understood. We develop a formal framework for majority-minority learning tasks, showing how standard training can favor majority groups and produce stereotypical predictors that neglect minority-specific features. Assuming population and variance imbalance, our analysis reveals three key findings: (i) the close proximity between "full-data" and stereotypical predictors, (ii) the dominance of a region where training the entire model tends to merely learn the majority traits, and (iii) a lower bound on the additional training required. Our results are illustrated through experiments in deep learning for tabular and image classification tasks.

## 1 Introduction

Imbalanced data are pervasive in machine learning, spanning rare-event detection, fraud, faults, medical anomalies, security, finance, and modern LLM pipelines with unequally represented subpopulations, see e.g., [12, 13] for some references. A sensitive case arises in fairness-related applications, where decisions apply to human beings [[ref]]. Addressing this issue is increasingly important notably under regulatory frameworks such as the European Union's AI Act, which emphasizes non-discrimination and risk mitigation.

In all these settings, the goal is to learn predictors that genuinely capture minority structure.

Our focus is on scenarios with two distinctive characteristics. First, the imbalance is typically significative, as we work directly with raw data without resampling or augmentation. Second, the imbalance is not corrected at the data level but addressed only through the training dynamics of gradient descent. An empirical fact, well known to practitioners, is that imbalance is not only preserved but often amplified by training: models initially align with the majority component and only later start to capture minority features. In the fairness literature, this is sometimes referred to as bias amplification [4, 10, 20, 21, 22]. Related simplicity-driven behaviors have been observed in representation learning [3, 9, 11, 17, 19].

This phenomenon has been documented since the 1990s in the class imbalance literature, see [1], which motivated numerous work and heuristics: data-level remedies (oversampling, under-sampling, synthetic examples [5, 18]), algorithm-level adjustments [8], cost-sensitive learning [7], focal and reshaped losses [14, 15]). With the advent of deep learning, the issue became even more acute, as high-capacity models and standard training budgets (a few hundred epochs) tend to privilege majority signals [12]. Despite this history, the mathematical mechanisms of imbalance amplification are still poorly understood geometrically, especially in nonlinear nonconvex regimes [16]. Yet, clarifying these mechanisms is essential for sensitive applications, as fairness, or for operational tasks like fraud or anomaly detection.

We develop a theoretical framework that explains why and how gradient-based training first produces stereotypical predictors, i.e. aligned merely with the majority, before entering a debiasing phase where minority features start to influence predictions. This two-phase picture subsumes quantitative

39th Conference on Neural Information Processing Systems (NeurIPS 2025).

notions such as the stereotype gap (distance between majority-only and full predictors) and explicit lower bounds on the debiasing overcost (the additional training required to move beyond stereotypes). Our objective focuses on understanding the geometry of the cost and the learning process: why would it lead to decisions that favor the majority group at the expense of the minority, as highlighted in [3]? We consider scenarios with significant imbalance in the training data, where the majority group is overrepresented in sample size and variability. In that case, under a typical training budget (e.g., 200–300 epochs in deep learning), models often amplify existing biases, producing stereotypical predictions disregarding minority populations. In contrast, prolonged and careful training on well-dimensioned architectures can detect minority-specific features more faithfully. We develop a mathematical framework to capture these phenomena and identify the core mechanisms behind bias amplification—specifically, population and variability imbalance, along with their geometric and dynamical implications.

**Contributions.** Our contributions are as follows.
— We first formalize the problem as a generic majority-minority learning task $\min L := L_1 + L_0$, with $L_0 \ll L_1$ using second-order differentiability domination. We prove that each critical point of $L$, which corresponds to a predictor, can be paired with a critical point of $L_1$, termed *stereotypical predictor*. We bound their distance: it is what we call the *stereotype gap*. It depends on a ratio measuring population and variance imbalance.
— The proximity of $L$ and $L_1$ implies that the region where minimizing $L$ is equivalent to minimizing $L_1$ (and vice versa) occupies nearly the entire parameter space. In linear regression, this results in a close overlap between $L_1$ training gradient path and the actual training path, illustrating how standard training neglects minority-specific characteristics.
— We prove that gradient descent may need a fairly long training time to merely identify stereotypical predictors, ignoring minority-specific aspects. Debiasing the model requires additional training; we derive a lower bound on this extra training duration. The corresponding ratio is called the *catch-up overcost ratio*; it quantifies the additional training time required to achieve unbiased predictions.
— Our approach also lays the groundwork for future work by introducing key notions and tools: stereotypical and representative predictors, majority–minority (adverse) zones, catch-up time, and debiasing time. These will be useful for analyzing other algorithms and for further study.
— We illustrate our theoretical findings through numerical experiments on several tabular and image classification tasks with deep neural networks (in the original paper accepted at Neurips `https://arxiv.org/abs/2505.13122` .

## 2 Predictions for majority-minority problems in machine learning

We first present our majority-minority scenario in as a minimization problem: $\min L := L_1 + L_0$. We aim at estimating the distance between a predictor obtained by minimizing the total loss $L$ and a neighboring majority-based predictor obtained by minimizing $L_1$. In practice, the latter may represent a biased or stereotyped view that a user holds about the underlying problem. We show that a small population and low variance for the minority group lead to proximity between the predictor and the majority-based predictor, making them difficult to distinguish.

**A majority-minority model.** We consider $n$ observations of a variable $Z := (X, Y) \in \mathbb{R}^d \times \mathbb{R}$ ($d > 0$) that can be divided into two groups following the values of a binary variable $A \in \{0, 1\}$. In our scenario, the data are unbalanced: there is a majority group $A = 1$ and a minority group $A = 0$, typically with $n_0 \ll n_1$. This heterogeneity, i.e., the variable $A$, may be unknown to the user.

Consider a collection of models or predictors $f_\theta : \mathbb{R}^d \mapsto \mathbb{R}$ indexed by parameters or weights $\theta \in \mathbb{R}^d$ that are learned by minimizing some empirical loss function over the learning set. Given a discrepancy measure $\ell : \mathbb{R}^2 \to \mathbb{R}_+$ we may define the total, majority and minority losses as: for $\theta \in \mathbb{R}^d$,

$$L(\theta) := \frac{1}{n} \sum_{i=1}^{n} \ell(f_\theta(X_i), Y_i) = \underbrace{\frac{1}{n} \sum_{\substack{i=1,\ldots,n \\ A_i=1}} \ell(f_\theta(X_i), Y_i)}_{:=L_1(\theta)} + \underbrace{\frac{1}{n} \sum_{\substack{i=1,\ldots,n \\ A_i=0}} \ell(f_\theta(X_i), Y_i)}_{:=L_0(\theta)}.$$

In the training phase of a learning process, the parameters are often computed through first order methods and thus eventually through vanishing gradients. Assuming both $\ell$ and $f_\theta$ are differentiable, we are thus led to consider equations of the form: $\nabla L(\theta) = 0, \quad \nabla L_j(\theta) = 0$, for $j \in \{0, 1\}$. In a

strongly imbalanced scenario, $L_0$ may become negligible with respect to $L_1$, so that the equations $\nabla L_1 = 0$ and $\nabla L = 0$ have very close solutions. On the other hand, this proximity does not prevent solutions to the equation $\nabla L_1(\theta) = 0$ from producing biased or stereotyped predictors as it ignores, by definition, the influence of data underlying $L_0$.

The aim of the following is to study this phenomenon and provide a set of assumptions for estimating the distance between full-data and stereotypical predictors.

The spirit of the following results is that $L_1$ corresponds to a majority behavior while $L_0$ is attached to minority features. In an analytical setting, it translates into a property of the type: $L_0$ is negligible w.r.t $L_1$ (see the assumptions below). We then aim at comparing $\operatorname{crit} L$ and $\operatorname{crit} L_1$; $\operatorname{argmin-loc} L$ and $\operatorname{argmin-loc} L_1$.

**Theorem 1** (Distances between critical points). *Consider two functions $L_1$ and $L_0$ from $\mathbb{R}^d$ to $\mathbb{R}$ that are two times continuously differentiable.*

*Assume some technical assumptions and that*

- *Strong Morse property: For all $\theta \in \mathbb{R}^d$,*

$$\|\nabla L_1(\theta)\| \leq c \implies \rho_{\min}(\nabla^2 L_1(\theta)) \geq \delta, \tag{1}$$

- *Bounds on the "minority loss":*

$$\sup_{\theta \in \mathbb{R}^d} \|\nabla L_0(\theta)\| \leq \tau, \quad \sup_{\theta \in \mathbb{R}^d} \rho_{\max}\left(\nabla^2 L_0(\theta)\right) \leq \tau. \tag{2}$$

*Then, for each $\widehat{\theta}_1 \in \operatorname{crit} L_1$ (resp. $\widehat{\theta} \in \operatorname{crit} L$) there exists a unique corresponding $\widehat{\theta} \in \operatorname{crit} L$ (resp. $\widehat{\theta}_1 \in \operatorname{crit} L_1$) such that*

$$\|\widehat{\theta}_1 - \widehat{\theta}\| \leq \frac{4\tau}{\delta}$$

*and $\widehat{\theta}, \widehat{\theta}_1$ have the same indexes, that is the same number of strictly negative eigenvalues of the Hessian matrices $\nabla^2 L_1(\widehat{\theta}_1)$ and $\nabla^2 L(\widehat{\theta})$.*

**Corollary 1** (Distances between critical and local minimizer sets). *In the context of Theorem 1, if $\operatorname{crit} L_1$ is non-empty, then $\operatorname{crit} L$ is non-empty and we have*

$$\operatorname{dist}_{\mathrm{H}}\left(\operatorname{crit} L_1, \operatorname{crit} L\right) \leq \frac{4\tau}{\delta}. \tag{3}$$

*Also, if $\operatorname{argmin-loc} L_1$ is non-empty then $\operatorname{argmin-loc} L$ is non-empty and we have*

$$\operatorname{dist}_{\mathrm{H}}\left(\operatorname{argmin-loc} L_1, \operatorname{argmin-loc} L\right) \leq \frac{4\tau}{\delta}. \tag{4}$$

*Finally, for each $\theta \in \operatorname{argmin-loc} L_1$, there is $\theta' \in \operatorname{argmin-loc} L$ such that the ball $B(\theta', \frac{6\tau}{\delta})$ contains $\theta$, and $L$ is $\delta/8$ strongly convex on this ball.*

A simple reading of Corollary 1 is that $L$ and $L_1$ are sufficiently close to share the same local Morse structure, the same 'geometry' so to say: they have the same number of local minimizers and, more generally, the same number of critical points for a given index, with, in addition, corresponding points lying at small distance from one another.

**A machine learning view: the representative and stereotypical predictions**: Let us interpret the above within a learning perspective. Under the premises of Theorem 1, we consider a machine learning model with loss $L : \theta \mapsto L(\theta)$ decomposed into a sum $L = L_1 + L_0$ where $L_1$ and $L_0$ respectively correspond to some majority and minority phenomena.

A critical point of $L$ is called a *representative prediction*, as it takes into account all available data encoded within $L$, i.e. both those in $L_1$ and $L_0$[1]. In the majority-minority model, the critical points of $L_1$ ignore data corresponding to the case when $A = 0$, we thus call them *stereotypical predictions*. The quantity $\operatorname{dist}_{\mathrm{H}}\left(\operatorname{crit} L, \operatorname{crit} L_1\right)$ is called the *stereotype gap*.

---

[1]It would be more natural to reserve that name for local minimizers, as those are generally obtained after training, but we do so for simplicity.

Roughly speaking Theorem 1 tells us, in particular, that each representative prediction corresponds to one and only one stereotypical prediction and that these predictions are close whenever the ratio

$$\Delta = \rho_{\max}\left(\nabla^2 L_0(\theta)\right) / \rho_{\min}(\nabla^2 L_1(\theta))$$

is uniformly small. This ratio is the key quantity that governs the stereotype gap.

The result is even more accurate, as Corollary 1 shows that the minimizers of $L$ and $L_1$ actually come by pairs as well, so that the stereotypical and representative predictors obtained in practice are "dangerously" close in a majority-minority scenario. As we will see through theoretical and numerical experiments, this renders the training phase delicate and potentially biased. Using the well-known fact that gradient descent converges to critical points in the Morse case, we may empirically estimate the stereotypical gaps and the associated "debiasing training time" in our imbalanced setting (see also the following sections).

Protocol (Table 1 opposite): find a stereotypical predictor $\widehat{\theta}_1$ via the gradient flow $-\nabla L_1$ with Kaiming random initialization. Initialize from this predictor $\widehat{\theta}_1$ and follow the flow of $-\nabla L$, with the guarantee (see Corollary 1) of reaching the corresponding representative predictor $\widehat{\theta}$. Use these values to estimate the gap $\mathrm{dist}_{\mathrm{H}}(\mathrm{crit}\, L, \mathrm{crit}\, L_1)$ via proxies like $\|\widehat{\theta} - \widehat{\theta}_1\|$, and to define a debiasing time from $\widehat{\theta}_1$ to its representative $\widehat{\theta}$ using gradient descent on $L$ with stopping criterion $\|\theta_{k+1} - \widehat{\theta}_1\| \geq 0.99\|\theta_k - \widehat{\theta}_1\|$.

Table 1: Stereotypical and representative predictions for imbalanced CIFAR-2 ($n_0/n \approx 3\%$) with ResNet 18. We report the average and standard deviation over 30 runs.

| Metric | Mean | $\pm$ Std |
|---|---|---|
| Debiasing time | 469 epochs | $\pm$ 9.4 |
| $\|\widehat{\theta} - \widehat{\theta}_1\|$ | 0.6723 | $\pm$ 0.0083 |
| $\|\widehat{\theta} - \widehat{\theta}_1\|_\infty$ | 0.0353 | $\pm$ 0.0047 |
| $\frac{\|\widehat{\theta} - \widehat{\theta}_1\|}{\|\widehat{\theta}\|}$ | 0.00602 | $\pm$ 0.00007 |

## 3 Learning unbalanced data with the gradient method

We study how gradient descent procedures may bias predictions in the sense that a "careless training" may yield a stereotypical predictor rather than a representative one. Gradient descent training on a $C^2$ loss $L$ is modeled through the ODE

$$\frac{d}{dt}\theta(t) = -\nabla L(\theta(t)) \text{ with } \theta(0) = \theta_{\mathrm{init}} \in \mathbb{R}^d. \tag{5}$$

For $C^2$ smooth losses $L = L_1 + L_0$, the *majority-training zone* is defined by

$$Z_{\mathrm{maj}} = \{\theta \in \mathbb{R}^d : \langle \nabla L(\theta), \nabla L_1(\theta)\rangle > 0\}.$$

In this region, descending along the gradient of $L$ also decreases $L_1$, and vice versa. In other words, $Z_{\mathrm{maj}}$ is a zone where training $L$ with gradient descent implies training the majority $L_1$. The *majority-adverse* zone is defined as

$$Z_{\mathrm{maj\text{-}adv}} = \{\theta \in \mathbb{R}^d : \langle \nabla L(\theta), \nabla L_1(\theta)\rangle \leq 0\} \text{ so that } \mathbb{R}^d \setminus Z_{\mathrm{maj}} = Z_{\mathrm{maj\text{-}adv}}. \tag{6}$$

We can similarly consider the minority-training and the minority-adverse zones. One easily sees that, under the Morse assumption, critical points of $L$ or $L_1$ lie in between $Z_{\mathrm{maj}}$ and $Z_{\mathrm{maj\text{-}adv}}$. In other words, the stereotypical and representative predictors lie on the boundary of $Z_{\mathrm{maj}}$.

We now establish two major facts: first, the majority zone is typically large, meaning that training the entire model often results in learning only the majority traits (see also the illustration of Figure 1); second, the majority-adverse zone promotes the training of the minority loss.

**Theorem 2** (Majority adverse zone). *Under Theorem 1 assumptions:*

$$Z_{\text{maj-adv}} \subset \bigcup_{\widehat{\theta}_1 \in \text{crit } L_1} B\left(\widehat{\theta}_1, \frac{2\tau}{\delta}\right).$$

**Remark 1** (On the majority-training zone size). Under the assumptions of Theorem 1, in the high dimensional regime the majority adverse zone has a volume lower than $O(4^{-d})$ —much lower in general as we have chosen a conservative bound. Note also that the stronger the imbalance, the more negligible it becomes.

**Lemma 1** (The majority adverse zone favors minority). *For $\theta \in Z_{\text{maj-adv}}$, we have*

$$\langle \nabla L(\theta), \nabla L_0(\theta) \rangle \geq 0.$$

*Thus a training trajectory $\theta : I \to \mathbb{R}^d$ evolving within $Z_{\text{maj-adv}}$ is such that $L_0(\theta(t))$ is non-increasing over the interval $I$.*

In other words, when the trajectory evolves within the majority-adverse zone, the dynamics learns minority features.

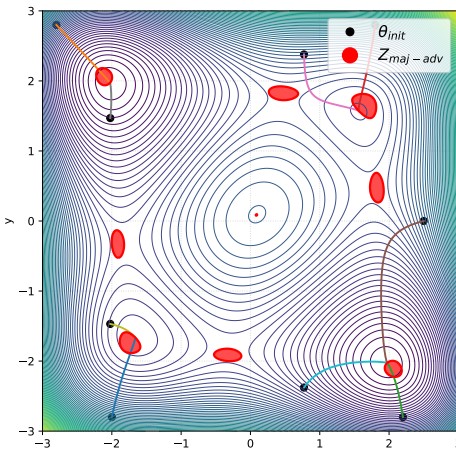

Figure 1: The majority region (white) covers nearly the entire space, while the majority-adverse region (red) is small. Training typically begins in white; hence majority features are learned during the initial phase, most of the path length (though not the time). The gradient trajectory then enters a red zone, where minority features are improved. The time spent there may be very long. These transient passages through red before convergence correspond to 'unlucky' curves.

Consider an "unlucky gradient curve" $t \mapsto \theta(t)$ solving (5) that effectively ignores the minority until it meets a majority predictor. On $[0, t_{\text{stereotype}}]$, the trajectory carries the initial condition $\theta_{\text{init}}$ to a critical point of $L_1$, viewed as a stereotype and denoted $\widehat{\theta}_{\text{stereotype}} := \theta(t_{\text{stereotype}})$. Up to time $t_{\text{stereotype}}$, it is 'as if' only $L_1$ were trained—the minority is entirely ignored. Thereafter, $\theta(t)$ moves toward a critical point of the full loss $L$, denoted $\widehat{\theta}$, which we interpret as a representative predictor. We prove in the full version that to debias the solution, it is necessary to continue training of the algorith. This extra time is interpreted as a catch-up time for the algorithm to detect the minority with an acceptable precision. It is a debiasing phase in which the algorithm progressively removes the bias it has itself created during the preliminary training phase. We prove that this time is long in comparison with $t_{\text{stereotype}}$.

## 4 Conclusion

Although our goals are primarily theoretical and future research should explore more refined training protocols, we can draw several conclusions supported by both theory and numerics—ours and the community's as well, see e.g., [1, 13]. These conclusions may also serve as recommendations for practitioners. Two key quantities emerge as critical in our study: the stereotype gap and the training duration. Additionally, we have empirical evidence that the model size may be a determining factor in achieving budget frugality.

— In a majority-minority scenario, population and variability imbalances are determining factors influencing the stereotype gap (Theorem 1 and the subsequent subsections). This gap, between stereotypes and representative predictors, can be very small in severely imbalanced cases.

— For convex or deep learning problems, gradient training generally leads to a "satisfying predictor" in the sense of a low-value loss $L$, see e.g., [2] or [6]. However, in our majority-minority scenario, the action of $L_0$ is generally almost indetectable, as shown in Figure 1, thus early stopping and under-dimensioned models are prone to produce stereotypes.

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
