# OpenReview forum: "when majority rules minority loses"
_EurIPS.cc/2025/Workshop/UPLB — UPLB2025_

### Official Review · Reviewer_MFo3 · 2025-10-24
**Solid Theoretical Insight into the Dynamics of Majority Bias**

**Rating:** 8
**Confidence:** 3

**Review:**

### Summary
The paper analyzes why models trained on imbalanced data first learn majority patterns (“stereotypes”) before gradually adjusting to minority patterns (“representative” predictors). It formalizes this with a decomposition of the loss $L = L\_1 + L\_0$, establishes proximity bounds between majority and overall optima (the “stereotype gap”), defines training regions favoring majority or minority learning, and quantifies the extra “catch-up” time needed for debiasing.

---

### Strengths
- Clear and elegant theoretical framing of bias dynamics in gradient-based training.
- Formal link between optimization geometry and fairness behavior (Theorem 1–2).
- Intuitive concepts (“stereotype gap”, “majority-adverse zone”) connecting theory and practice.
- The exposition succeeds in conveying the main ideas (results of a rigorous mathematical analysis) clearly and intuitively.

---

### Weaknesses / Concerns
- Assumptions (Morse property, bounded curvature ratio) are mathematically clean but arguably unrealistic for deep networks.
- While the empirical validation is mentioned, it is not shown. Given the theoretical nature of the paper and space constraints, this is understandable; however, attaching complementary results in an appendix would have been valuable.
- Similarly, the extensive proofs are omitted. The focus on intuition works well, but including detailed proofs in an appendix would enhance clarity and traceability.
- Some definitions and notations are heavy, making Sections 3–4 difficult to follow without rereading.
- **Minor:** placeholder citations (e.g., “[ref]”) and few typos (e.g. "algorith", "indetectable")

---

### Significance
The work presents an original and rigorous framing of majority–minority learning dynamics. Beyond theoretical insight, the framework provides a quantitative basis to estimate the training time required to exit the majority-bias regime through the definition of the “catch-up overcost”. This offers practical value for understanding and controlling the transition from majority fitting to effective minority learning.

---

### Recommendations for Improvement
- Clarify the realism of assumptions and provide diagnostics to estimate key parameters $(\delta, \tau, \Delta)$ empirically.
- Add a supplementary appendix with empirical illustrations and full proofs.
- Refine exposition and remove placeholder references.
- Release code or pseudo-code for reproducibility.

---

### Questions for Authors
- Why must the parameter vector $\theta$ have the same dimensionality as the input $X$, as mentioned in the second half of page 2? Is a linear form for the model assumed? Please clarify this point and specify on which model assumptions the presented results rely.
- Theorem 1 is formulated on two arbitrary functions, assuming only the Morse property on one and bounded gradient and Hessian norms on the other. The result is striking given the generality of these assumptions. While you already provide some intuition on the implications of the theorem, could you elaborate further on how or why such limited hypotheses are sufficient for Theorem 1 to hold in full generality?

---

### Overall Assessment
An original contribution on bias dynamics in imbalanced learning. Strong theoretical insight, moderate empirical support. Supported by complementary material, it could become a notable reference in fairness-aware optimization.

---

### Decision · Program_Chairs · 2025-11-03

Accept (Oral)